# Exploring the Potential Applications of the Noninvasive Reporter Gene *RUBY* in Plant Genetic Transformation

Jingjing Yu [1,2], Shiling Deng [1,2], Han Huang [1,2], Jinhui Mo [1,2], Zeng-Fu Xu [1,2,*] and Yi Wang [1,2,*]

1. State Key Laboratory for Conservation and Utilization of Subtropical Agro-Bioresources, College of Forestry, Guangxi University, Nanning 530000, China
2. Key Laboratory of National Forestry and Grassland Administration on Cultivation of Fast-Growing Timber in Central South China, College of Forestry, Guangxi University, Nanning 530000, China
* Correspondence: zfxu@gxu.edu.cn (Z.-F.X.); wangyi1984@gxu.edu.cn (Y.W.)

**Abstract:** Betalains can be conveniently observed and quantified and, accordingly, have the potential as naked-eye visual screening reporters during plant transformation. *RUBY* is a new reporter system that uses "2A" peptides to fuse three key genes, *CYP76AD1*, *DODA*, and *glucosyl transferase*, for betalain biosynthesis, and has been successfully used for transformation of rice, *Arabidopsis*, and cotton, but its potential applications in the genetic transformation of various other plant species remain to be verified. In this study, *RUBY* was transferred into the hairy roots of *Plukenetia volubilis* and *Nicotiana benthamiana*, and was transferred into *Arabidopsis* by the floral-dip method. The expression levels of *CYP76AD1*, *DODA*, and *glucosyl transferase* were detected by RT−PCR and RT−qPCR, the relationship between the expression level of *RUBY* and red coloration was analyzed, and the genetic stability of *RUBY* in transgenic *Arabidopsis* was studied. The results showed that the expression of *RUBY* could reconstruct the betalain biosynthesis pathway in the hairy roots of *P. volubilis*, *N. benthamiana*, and *Arabidopsis* plants, indicating that it has the potential for versatile use across species. As a reporter, betalain did not affect callus induction, plant regeneration, development, or fertility. However, when used in plant transformation for observation and visual screening, it needed to accumulate to a certain extent to show red coloration, and it was positively correlated with gene expression. In general, *RUBY* is a convenient reporter for plant transformation, and has no obvious side effects during plant growth and development. However, the potential application of *RUBY* for visual screening is highly determined by the expression level, and further improvement is needed.

**Keywords:** betalain; *RUBY*; *Plukenetia volubilis* L.; hairy root; genetic transformation

## 1. Introduction

Betalains, anthocyanins, and carotenoids are three major natural pigments in plants. Of the three pigments, betalains have the simplest biosynthesis pathway: only *CYP76AD*, *DODA*, and *glucosyl transferase* are needed to synthesize betalain in four steps of catalysis, using tyrosine as the substrate [1–3]. Previous studies have shown that betalain could be used as a noninvasive reporter, which means that unlike the reporter gene *GUS*, it does not require staining and sacrificing of plant tissues to track the process of arbuscular mycorrhizal fungal colonization by *Rhizophagus irregularis* in *Medicago truncatula* and *N. benthamiana* roots [4,5]. The expression of *CYP76AD*, *DODA*, and *glucosyl transferase* driven by tissue-specific promoters led to the accumulation of betalain in specific tissues, such as fruits and endosperm [6,7]. When replaced with inducible promoters, such as *AtYUC4* and *DR5*, betalain was induced and accumulated where auxin was distributed [8]. Compared with those of *MYB* and the basic *HELIX-LOOP-HELIX* (*bHLH*) transcription factors that regulate anthocyanin synthesis, the biosynthesis of betalain is more universal across species, and it has now been reconstructed by genetic engineering in multiple betalain-free species,

such as *A. thaliana*, *Nicotiana tabacum*, *Petunia hybrida*, *Oryza sativa*, *Solanum lycopersicum*, *S. tuberosum*, and *S. melongena* [4,6,9–11]. In angiosperms, betalains and anthocyanins might be mutually exclusive [12], but the overexpression of *CYP76AD*, *DODA*, and *glucosyl transferase* in tomatoes resulted in the production of both anthocyanins and betalains in fruits, which presented uniform coloration distinct from WT fruit pigmentation [11,13]; hence, betalain is expected to be a more efficient candidate reporter than anthocyanins [12].

*RUBY* is a naked-eye visual reporter based on the characteristics of betalain. It uses a 2A peptide to link *CYP76AD*, *DODA*, and *glucosyl transferase* into a fusion gene. With the help of "ribosome jumping" and "self-cleavage technology", it generates three enzymes that catalyze the biosynthesis of betalain [8]. *Arabidopsis* and rice transformed with *RUBY* displayed a prominent red color, thus allowing the differentiation of transgenic callus or plants from wild-type, and enabling the efficient screening of positive transformants in genetic transformation [8,14–18]. However, the generality and stability of *RUBY* in plants remain to be verified.

*P. volubilis* is a woody vine oil crop whose seeds contain approximately 45%–50% lipids and have high economic value [19–24]. In recent years, using *Agrobacterium rhizogenes* to induce transgenic hairy roots has been successful in a variety of woody plants [25–28]. Those studies provide new insight into the genetic transformation of recalcitrant woody plants such as *P. volubilis*. To assess the potential applications of *RUBY* and increase the efficiency of the genetic transformation of *P. volubilis*, the 346 bp *Cauliflower mosaic virus* (*CaMV*) *35S* promoter was designed to drive *RUBY* ectopic expression in this study. Transgenic hairy roots of *P. volubilis* were obtained by *A. rhizogenes*-mediated transformation, and red transgenic callus was successfully induced. At the same time, *RUBY* was transferred into the model plants *N. benthamiana* and *Arabidopsis* to explore the relationship between *RUBY*'s red coloration and its expression, and to analyze the expression stability of *RUBY* in transgenic *Arabidopsis*. These results provided a reference for the screening of reporter genes for the majority of woody plants that currently lack an established and efficient genetic transformation system.

## 2. Materials and Methods

### 2.1. Plant Materials and Growth Conditions

The seeds of *P. volubilis* and *N. benthamiana* were surface-sterilized with 0.1% mercuric chloride for 5–10 min and then washed five times with sterile water. Subsequently, the seeds of *P. volubilis* and *N. benthamiana* were sown on 1/2 woody plant medium (WPM) and 1/2 Murashige and Skoog (MS) solid medium, respectively, and cultured in a growth chamber at 100–150 $\mu$mol/m$^{-2}$·s$^{-1}$ under a 16 h light/8 h darkness cycle at 24–26 °C. The stem segments and leaves of sterile *P. volubilis* and *N. benthamiana* seedlings were taken as explants for genetic transformation. Sterilized seeds of Col-0 were sown on 1/2 MS solid medium and grown for 14 days in greenhouse, and then seedlings were transplanted to soil to grow until bolting for floral-dip transformation.

### 2.2. Construction of the 35S$_{346 bp}$::RUBY Expression Vector

The pRN114 plasmid [29] was used as a template, and the specific primer was designed to amplify the *CaMV 35S* promoter with a length of 346 bp without an enhancer [30], named 35S$_{346 bp}$. This fragment was ligated into *DR5::RUBY* [8], which had been digested by *Sma* I and *Pst* I to construct 35S$_{346 bp}$::RUBY by an In-fusion PCR Cloning Kit (Vazyme, Code No. C112−01, China). The recombinant plasmid 35S$_{346 bp}$::RUBY was verified by sequencing and transferred into *A. tumefaciens* GV3101 and *A. rhizogenes* K599 by the heat shock method for genetic transformation [31].

### 2.3. Genetic Transformation of RUBY

Single colonies of *A. rhizogenes* strain K599 and *A. tumefaciens* strain GV3101 harboring 35S$_{346 bp}$::RUBY were isolated from plates, inoculated in 10 mL of LB liquid medium with 50 mg·L$^{-1}$ kanamycin and 20 mg·L$^{-1}$ rifampicin, with shaking (180 rpm) at 28 °C for

12–14 h, transferred into 200 mL LB liquid medium to incubate until the $A_{600\ nm}$ was 0.6, and centrifuged at 4000 rpm for 5 min to collect the bacteria. The cells were resuspended in 200 mL 1/2 MS liquid medium (pH = 5.8) containing 100 μM acetosyringone to an $A_{600\ nm}$ of 0.4–0.8, which was prepared to infect explants of *P. volubilis* and *N. benthamiana*. When infecting *P. volubilis* by *A. rhizogenes* strain K599, the hypocotyls of 10-day-old sterile seedlings or young stem segments located at the top of 2–3 cm on 40-day-old sterile seedlings were selected as explants. The induction of *P. volubilis* transgenic hairy roots *in planta* was carried out by a "one-step *A. rhizogenes*-mediated" (ARM) method with some modifications [32], and three chimeric plants were obtained. When infecting *N. benthamiana* by *A. tumefaciens* strain GV3101, the leaves of 30-day-old sterile seedlings were selected as explants, and 29 *RUBY* transgenic plants were obtained.

### 2.4. Genetic Transformation of 35S_{346 bp}::RUBY Arabidopsis

*Arabidopsis* transformation was performed by the floral-dip method [33]. The seeds of transformed *Arabidopsis* were sterilized by chlorine gas and sown evenly on the screening medium (25 mg·L$^{-1}$ hygromycin). The genomic DNA from resistant *Arabidopsis* plants was extracted with the modified SDS-alkali lysis method [34]. A pair of specific primers, *RUBY−F* and *RUBY−R* (Table S1), were designed to amplify full-length *RUBY* to identify transgenic *Arabidopsis*.

### 2.5. RNA Extraction, RT−PCR, and RT−qPCR Analysis

The total RNA of transgenic *Arabidopsis*, transgenic hairy roots of *P. volubilis*, and *N. benthamiana* was extracted by TRIzol reagent (Thermo, Code No.15596026, Massachusetts, USA). The synthesis of cDNA from RNA was performed by using a HiScript$^{®}$ III All-In-One RT MasterMix kit (Vazyme, Code No. R333−01, Nanjing, China) and diluted 10 times as a template. Specific primers (Supplementary Table S1) were designed to detect the expression of *CYP76AD1*, *DODA*, and *glucosyl transferase* by RT−PCR and RT−qPCR. Actin homologs in *Arabidopsis*, *P. volubilis*, and *N. benthamiana* were used as reference RNA. The SYBR green fluorescence RT−qPCR analysis was performed with three biological replicates for each *N. benthamiana* transgenic hairy root and transgenic *Arabidopsis* sample, whereas samples from *P. volubilis* transgenic hairy roots were analyzed with three technical replicates. The relative expression of genes was calculated using the $2^{-\Delta\Delta Ct}$ method [35]. Graphs were drawn by using GraphPad Prism 8.0.

### 2.6. Extraction and Quantitative Determination of Pigment of RUBY Transgenic Plants and Tissues

Two grams of fresh *RUBY* transgenic callus of *P. volubilis* induced from red hairy roots and the pulp of red dragon fruit were weighed and stirred in 20 mL of 75% ethanol solution (pH = 5.2) as a solvent for 50 min. After stirring, the samples were centrifuged at 4000× *g* at 25 °C for 5 min. Two kinds of the obtained extracts were analyzed with a Nanodrop spectrophotometer. The absorption spectra of the two pigment extracts were drawn with GraphPad Prism 8.0 [36]. The 14-day-old *Arabidopsis* seedlings of lines #3 and #6 were classified into fully colored (cotyledons, hypocotyls, and roots were all of red coloration) and semicolored seedlings (only hypocotyls or roots were of red coloration). The extracts of each kind of seedling were pipetted into the wells of a 96-well microtiter plate, and the absorbance was measured at $A_{538\ nm}$ and $A_{400\ nm}$ with a microplate reader. The contents of red pigments were calculated and converted into the corresponding betalain mass fraction [37].

## 3. Results

### 3.1. Ectopic Expression of RUBY Reconstructed the Betalain Biosynthesis Pathway in P. volubilis

To explore the potential applications of *RUBY* in the genetic transformation of woody plants, we transformed the hypocotyls of *P. volubilis* with *DR5::RUBY* by using *A. rhizogenes* strain K599. We found that only a low rate of *RUBY* transgenic hairy roots was reddish. *DR5::RUBY* was transformed into 20 explants, and its transgenic hairy roots

were induced after approximately 3 weeks. Among them, a total of 30 hairy roots were induced from 14 explants, only 6 of which were red at the root tips, and the rest were white (Figure 1a). This might be because the expression of the *DR5* promoter was affected by auxin concentration at the root tips, as previously reported [8,38,39]. Subsequently, to enhance the expression level of *RUBY*, the *DR5* promoter was replaced by the 346 bp *CaMV 35S* promoter that did not contain an enhancer [30], which was widely used to drive the expression of selectable markers on the pCAMBIA series of vectors. Next, $35S_{346\,bp}::RUBY$ was transferred into young stems of *P. volubilis* mediated by *A. rhizogenes* strain K599. The red transgenic hairy roots were induced (Figure 1b), but the problem of low rates of red coloration remained (Supplementary Figure S1). We extracted pigments from *RUBY* transgenic callus (Figure 1c) and used red dragon fruit as a control, since red dragon fruit was commonly known as a primary source of betalain [37,40]. Both extracts were bright red (Figure 1d), and their absorption spectra measured with spectrophotometers showed that they had the same maximum absorption wavelength at $A_{538}$ nm (Figure 1e), which indicates that the pigments from *RUBY* transgenic callus were most likely the same as those in red dragon fruit [37,40]. At the same time, the absorption peak of the pigment extracted from *RUBY* callus of *P. volubilis* was higher than that of red dragon fruit at 400–460 nm, which may contain other pigments. Previous studies have found a similar phenomenon, such as in tomatoes [13]. These results showed that the transformation of *RUBY* could reconstruct the betalain biosynthesis pathway in the hairy roots of *P. volubilis*.

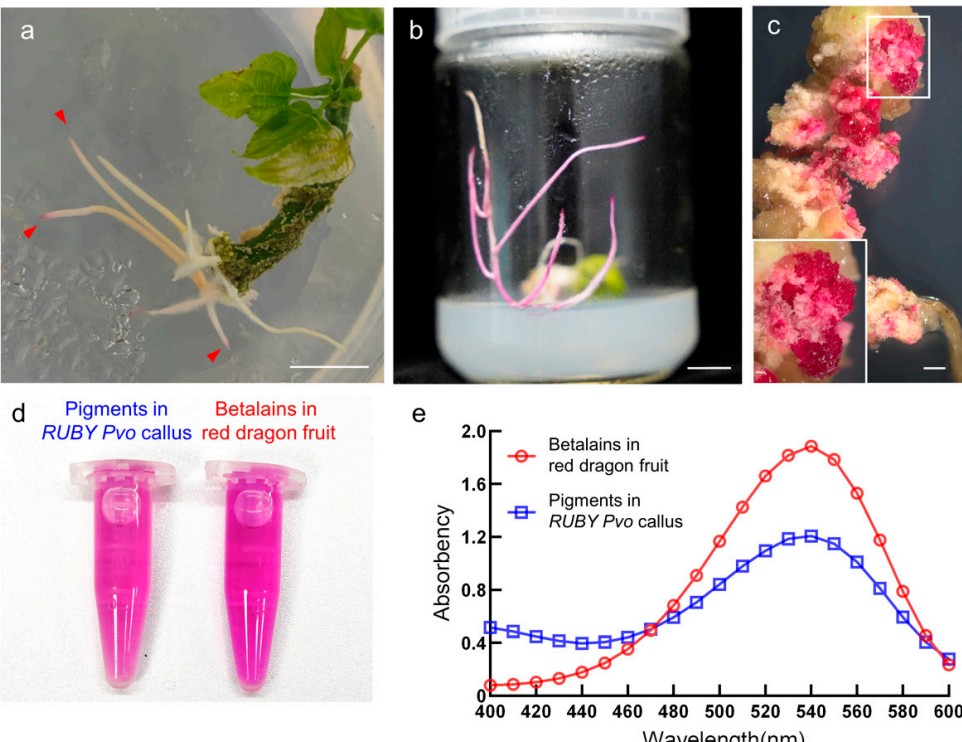

**Figure 1.** Induction of *RUBY* transgenic hairy roots and callus of *P. volubilis* and pigment identification. (**a**) *DR5::RUBY* transgenic hairy roots, the red triangles indicate the *RUBY*-colored root tips. (**b**) $35S_{346\,bp}::RUBY$ transgenic hairy roots. (**c**) Red callus induced from *RUBY* transgenic hairy roots of *P. volubilis*. The inset in the lower left corner is an enlarged image of the white box in the upper right corner, showing red callus induced from *RUBY* transgenic hairy roots. (**d**) Extracted pigment from *RUBY* callus of *P. volubilis* and red dragon fruit. (**e**) Absorption spectrum of the pigment from transgenic *RUBY* callus of *P. volubilis* and red dragon fruit (scale bar = 1 cm).

*3.2. Ectopic Expression of RUBY Has No Adverse Effects on the Growth and Development of Plants*

To verify whether continuous expression of *RUBY* and consumption of tyrosine would affect the normal growth and development of plants, we first used *RUBY* transgenic red hairy roots of *P. volubilis* as explants to induce a large number of red calli on WPM medium containing 0.2 mg·L$^{-1}$ IBA and 2 mg·L$^{-1}$ 6-BA (Figure 2a). Second, three chimeric *P. volubilis* plants containing red hairy roots were obtained by the ARM (*A. rhizogenes-*mediated) method [32] (Figure 2b), and the plants grew normally without any abnormalities under the growth conditions tested here. Third, $35S_{346\ bp}::RUBY$ was transferred into Col-0, and 10 resistant seedlings were obtained. PCR detection showed that nine of them were positive (Supplementary Figure S2), and #1–#7 *RUBY* transgenic *Arabidopsis* were selected for further analysis. The siliques of those transgenic *Arabidopsis* plants were dissected, and we did not observe seed abortion (Figure 2c). Finally, the leaves of *N. benthamiana* were used to perform the genetic transformation with *A. tumefaciens* strain GV3101, and approximately 29 red *RUBY* transgenic shoots were obtained (Supplementary Figure S3). These transgenic shoots rooted to form 13 complete plants, which were grown in soil (Figures 3a–c and S4). Compared with the wild-type, *RUBY* transgenic *N. benthamiana* plants and their leaves, stems, flowers, fruits, and seeds showed no developmental defects except for the red coloration, due to the accumulation of betalain (Figure 3d–k). These experimental results indicate that the continuous consumption of tyrosine by overexpression of *RUBY* may not adversely affect plant growth, regeneration, or reproduction.

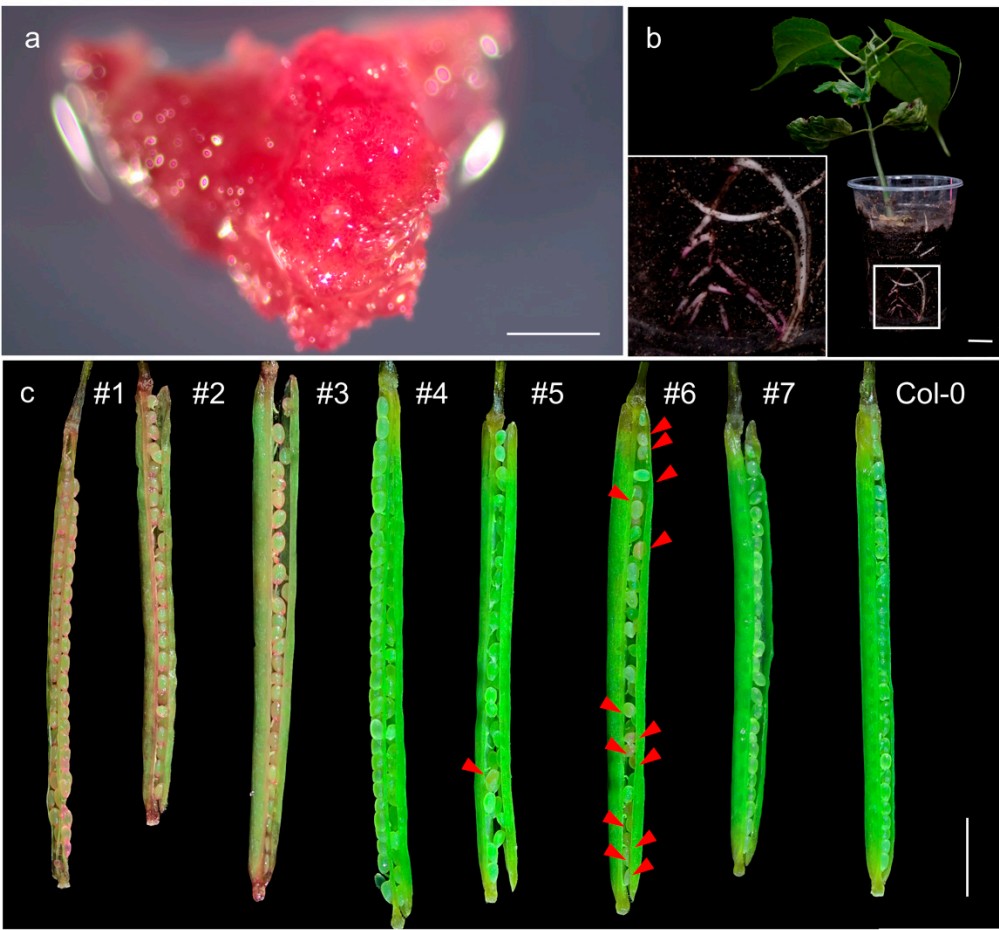

**Figure 2.** Nondestructive analysis of ectopic expression of *RUBY* in *P. volubilis* and *Arabidopsis*. (**a**) Red transgenic callus induced from *RUBY* transgenic hairy roots of *P. volubilis* as explants. (**b**) *P. volubilis* chimeric plants. (**c**) Dissected silique of *RUBY* transgenic *Arabidopsis* lines #1–#7, the red triangles indicate red coloration (scale bars in (**a,b**) = 1 cm, scale bar in (**c**) = 200 μm).

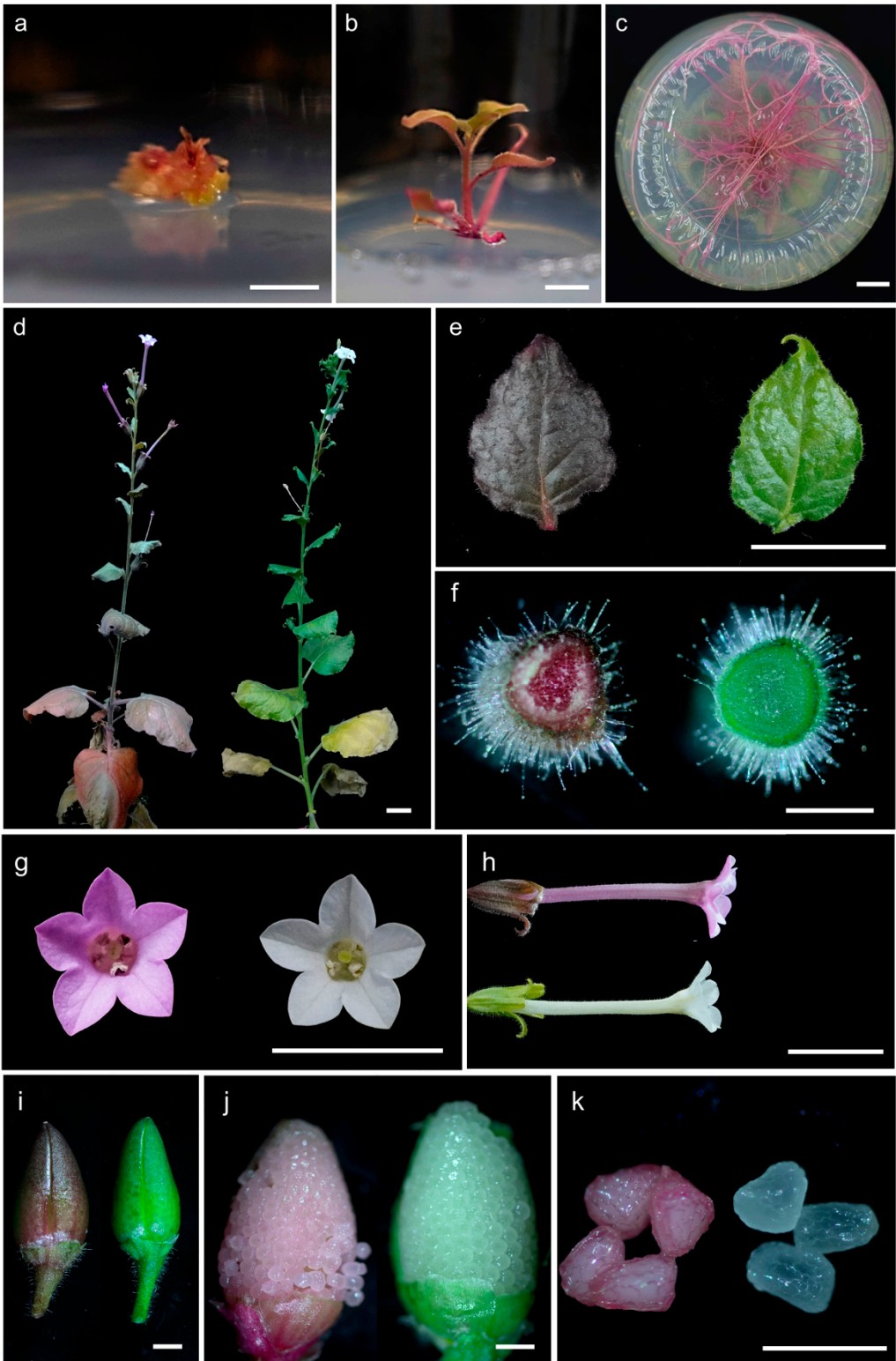

**Figure 3.** The phenotype analysis of *RUBY* transgenic *N. benthamiana* plants. (**a**,**b**) *RUBY* transgenic adventitious bud regenerated from *N. benthamiana* leaf. (**c**) *RUBY* transgenic *N. benthamiana* plant induced roots. (**d**) The whole plant morphology of *RUBY* transgenic *N. benthamiana* (left) and WT (right), the same below. The leaf (**e**), stem (**f**), flower (**g**,**h**), and fruit (**i**) morphology of *RUBY* transgenic *N. benthamiana* and WT. Peeled semimature fruit (**j**) and seeds (**k**). (Scale bars in (**a**–**e**,**g**,**h**) = 1 cm, scale bar in (**f**,**i**–**k**) = 1 mm).

### 3.3. RUBY Transgenic Hairy Roots and Plants Showed Low Rates of Red Coloration

To explore whether the low rate of red coloration of *RUBY* was caused by species specificity, we used *A. rhizogenes* K599 harboring $35S_{346\,bp}::RUBY$ to infect the leaves of *N. benthamiana* and successfully induced a large number of hairy roots (Figure 4a,b), but the problem of a low rate of red coloration remained. We also found that some white hairy roots of *N. benthamiana* also grew normally in the selection medium with 15 mg·L$^{-1}$ hygromycin. Therefore, we randomly selected nine white hairy roots and performed RT-PCR. After 25 cycles of PCR amplification, the transcripts of *CYP76AD1*, *DODA*, and *glucosyl transferase* were not detected in those hairy roots, but when the number of PCR cycles was increased to 30, three gene transcripts were detected except in the roots of WT, #1, and #2 (Figure 4c). It has been reported that when *A. rhizogenes* transferred the target gene into hairy roots, it simultaneously transferred auxin genes, such as *rol A* on the *Ri* plasmid, into the recipient cell genome, resulting in abnormal concentrations of auxin [41–43].

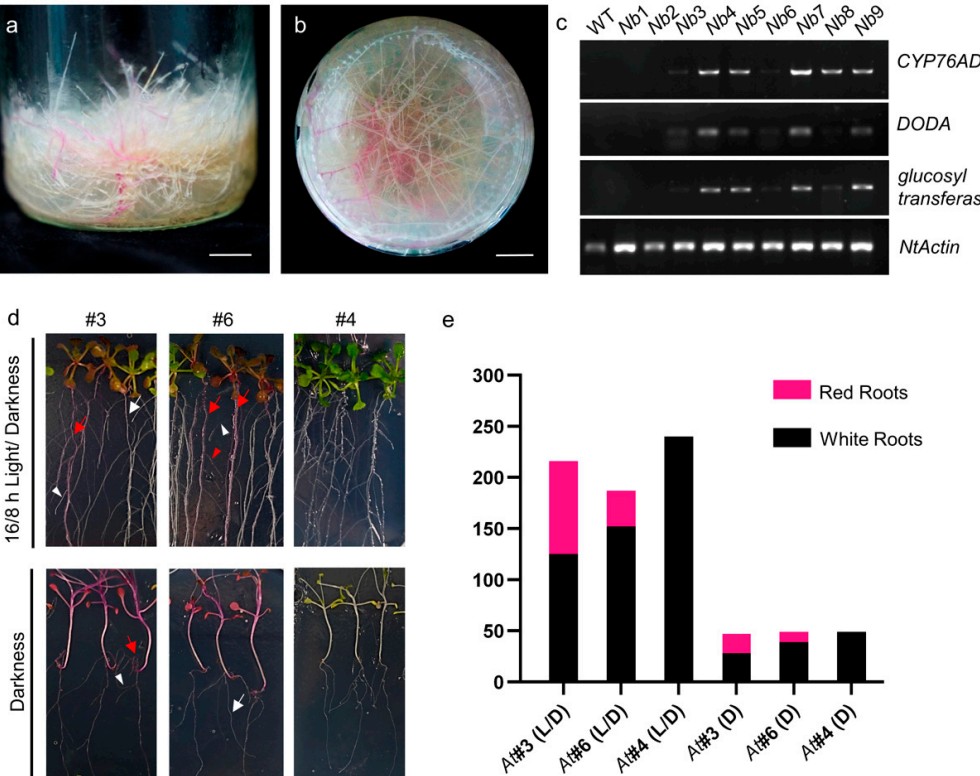

**Figure 4.** Analysis of the red coloration rate of overexpressed *RUBY* in *N. benthamiana* hairy roots and *Arabidopsis*. (**a,b**) *RUBY* transgenic hairy roots induced from *N. benthamiana* leaves, (**a**) side view of the bottle, (**b**) bottom view of the bottle. (**c**) RT−PCR detecting the expression levels of *CYP76AD1*, *DODA*, and *glucosyl transferase* in nine noncolored hairy roots of *N. benthamiana*. The roots of wild-type were used as controls. "*Nb*" indicates templates from *N. benthamiana*. (**d**) Root growth and red coloration rate of T2 seedlings of *RUBY* transgenic *Arabidopsis* lines under 16/8 h (light/darkness) and darkness (red arrows or triangles indicate colored roots, the white triangles indicate noncolored roots). (**e**) The statistics of root red coloration rate of *RUBY* transgenic *Arabidopsis* lines #3 (fully colored), #4 (noncolored), and #6 (semicolored). "*At*" indicates samples of *A. thaliana*.

To further rule out the possibility that the low rate of red coloration was caused by the characteristics of transgenic hairy roots, we observed nine lines of $35S_{346\,bp}::RUBY$ transgenic *Arabidopsis* generated with *A. tumefaciens* and found that only 33.3% were colored. We selected seeds of lines #3 (fully colored), #4 (noncolored), and #6 (semicolored) from transgenic *Arabidopsis* and sowed them on vertical plates that contained 25 mg·L$^{-1}$ hygromycin. After 30 days of growth, 15 fully colored plants of *RUBY* transgenic *Arabidopsis*

line #3 had a total of 216 roots (including taproots and primary lateral roots), of which only 42.1% were red roots, and the remaining 57.9% of roots were white or almost white. In the semicolored *RUBY* transgenic *Arabidopsis* line #6, only 5 fully colored plants and the remaining 10 plants were semicolored or noncolored. These plants grew a total of 187 roots, among which only 18.7% of roots were red, and the remaining 81.3% of roots were white. The rate of red coloration of roots was distinctly lower than that in line #3. A total of 240 roots grew from the 15 plants of noncolored line #4, all of which were white. The elongation and number of roots under dark conditions were much lower than those under 16 h light/8 h dark cycles. Nineteen red roots and 28 white roots grew from 15 plants of *RUBY* transgenic *Arabidopsis* line #3, while 10 red roots and 39 white roots grew from line #6, and 49 white roots grew from line #4 (Figure 4d,e). The rates of red coloration of *RUBY* transgenic *Arabidopsis* were low under both planting conditions, which was consistent with the phenomenon observed in *P. volubilis* and *N. benthamiana*. The above experiments showed that *RUBY*, as a "noninvasive" naked-eye visualization reporter, did have advantages in genetic transformation, but it had the problem of a low rate of red coloration in transgenic hairy roots and plants, which had not been previously reported.

*3.4. The Expression Level of RUBY Is Positively Correlated with Red Coloration*

We found that one of the *RUBY* transgenic hairy roots obtained from *P. volubilis* turned from white to red during subculture: the primary root was white, but the whole or part of the secondary lateral roots were red (Supplementary Figure S5). To explore the reason for the different red coloration in this *RUBY* transgenic hairy root, we divided it into seven regions for RT−PCR detection. The transcripts of *CYP76AD1*, *DODA*, and *glucosyl transferase* were detected in regions no. 1–4 and region no. 6 after 25 cycles of PCR amplification, but region no. 5 and region no. 7 had lower levels (Figure 5a).

Among the T1 generation of *RUBY* transgenic *Arabidopsis*, only lines #1, #2, and #3 had red siliques and seeds. The siliques of line #6 were as green as those of the wild-type, but some seeds were red (Figure 2c). In addition, we also found that the fully colored line #3 had white inflorescence and green siliques that were consistent with those of the wild-type at a later stage of growth (Supplementary Figure S6). We selected *RUBY* transgenic *Arabidopsis* lines #1–#6, and inflorescences on line #6 for RT−PCR detection, and the results showed that *CYP76AD1*, *DODA*, and *glucosyl transferase* were expressed in all transgenic lines: lines #1, #2, and #3 had high expression levels, while lines #4, #5, and #6 had lower expression levels; and the same results in inflorescences of line #6 (Figure 5b).

To further quantitatively analyze the relationship between the expression of *RUBY* genes and red coloration, we first detected the expression levels of *CYP76AD1*, *DODA*, and *glucosyl transferase* in the transgenic hairy roots of *P. volubilis* and transgenic *Arabidopsis* inflorescences by RT−qPCR. The expression of the three genes mentioned above could be detected in *RUBY* transgenic hairy roots of *P. volubilis* in each region, and the expression levels of those genes in regions no. 1–3 were higher than those in regions no. 5–7 (Figure 5c). However, the expression levels of those genes in region no. 2 had no significant difference from that of region no. 5. In *Arabidopsis*, the expression levels of the three genes in colored lines #1, #2, and #3 were higher than those in the noncolored lines, while in the white inflorescence of line #3, the expression levels of those genes were as low as those in the noncolored lines (Figure 5d). The expression level in line #3 was the lowest among those of the colored lines, and was significantly different from that of line #5, which had the highest expression among the noncolored lines.

In addition, we used RT−qPCR to detect the expression of *RUBY* in nine white transgenic hairy roots of *N. benthamiana* (Figure 5e). The results showed that three genes comprising *RUBY* were detected in all roots except no. 2, and the expression abundance was less than 0.08 times that of *Actin* (Figure 5e). The red coloration of nine white transgenic hairy roots could not be observed with the naked eye, probably due to the relatively low levels of expression. Based on the results of the above experiments, we speculated that the

high expression of *RUBY* would lead to the accumulation of betalain and red coloration in transgenic plants.

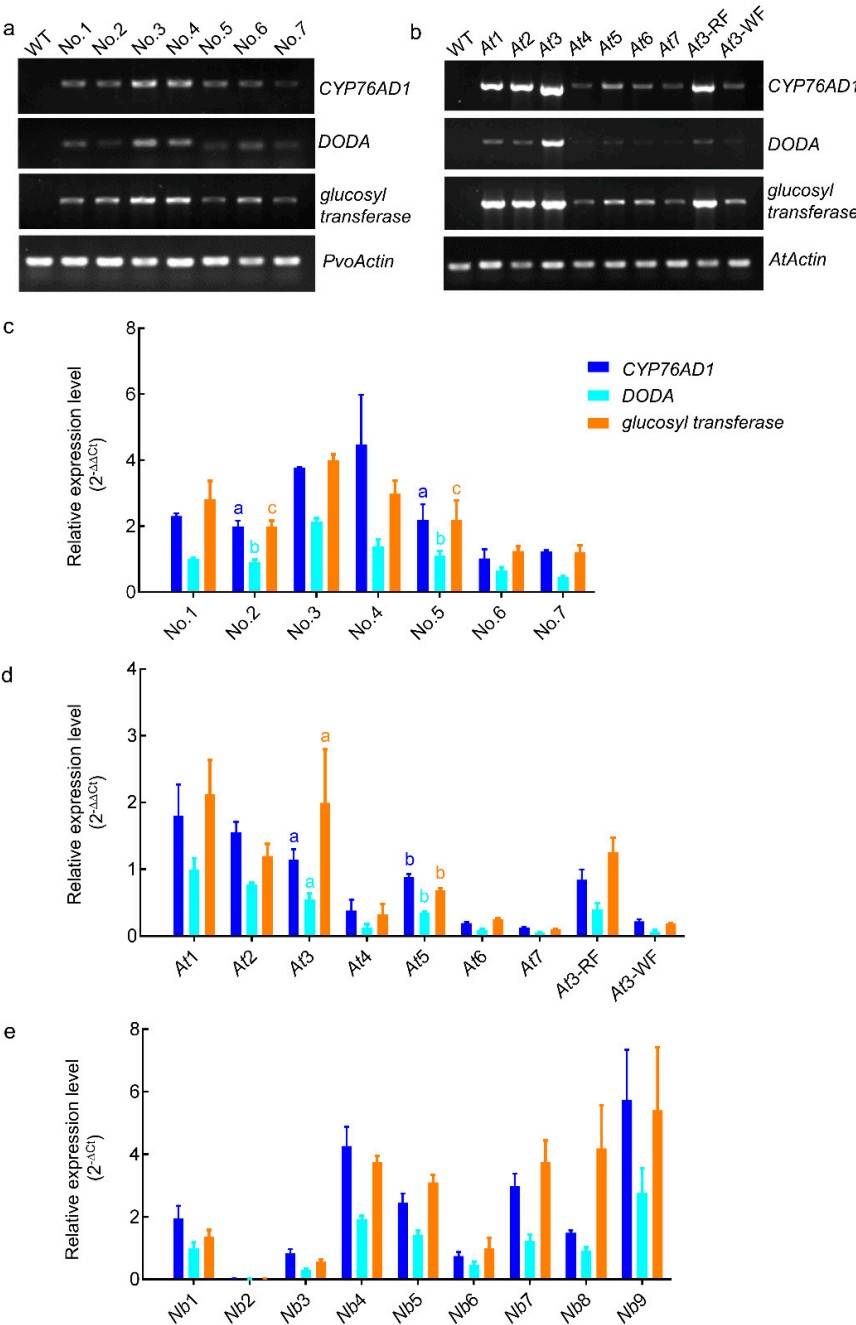

**Figure 5.** The expression analysis of *CYP76AD1*, *DODA*, and *glucosyl transferase* in transgenic hairy roots and plants. (**a**) RT−PCR detection of the expression of three genes in different regions of *RUBY* transgenic hairy root of *P. volubilis*. No. 1−7 are different regions of *RUBY* transgenic of *P. volubilis* hairy roots, and WT is the wild-type root. (**b**) RT−PCR detection of the expression of three genes in *RUBY* transgenic *Arabidopsis*. *At*1−*At*3 are fully colored lines, and *At*4−*At*7 are noncolored lines. *At*3−RF and *At*3−WF were red inflorescence and white inflorescence of line #3, respectively. (**c**–**e**) RT−PCR detection of three gene expression levels in *RUBY* transgenic hairy roots of *P. volubilis* (**c**) and *N. benthamiana* (**e**), and *RUBY* transgenic *Arabidopsis* lines #1–#7 and red and white inflorescence in line #3 (**d**). Vertical bars indicate standard errors of means (*n* = 3), different letters indicate significant differences at *p* < 0.05 (Fisher's LSD test) between the corresponding values of "No.2" and "No.5"and "*At*3" and "*At*5", respectively.

*3.5. Instability of RUBY Expression in Transgenic Arabidopsis*

According to a previous study, the fusion gene design of *RUBY* might cause abnormal transcription events, and there might be unknown risks in inheritance in the use of *RUBY* as a reporter [3]. We counted and calculated the rates of red coloration of *Arabidopsis* T2 seedlings in three fully colored lines (#1, #2, and #3) and three noncolored lines (#4, #5, and #6) (Supplementary Figure S7), and the results are shown in Table 1. The results showed that the ratios of surviving seedlings to dead seedlings from #1–#6 lines were all close to 3:1. It suggests that a single *RUBY* transgene could be stably passed on to the next generation according to Mendelian inheritance. However, the offspring of the fully colored lines #1, #2, and #3 had a considerable amount of semicolored seedlings with only the hypocotyls or roots colored (Figure 6a), which accounted for approximately 6.36% and 11.44% of lines #1 and #2, respectively; line #3 had simultaneous red and white inflorescences in the T1 generation, and the proportion of semicolored seedlings in the T2 generation was slightly higher (approximately 19.1%), but not significantly different from that of lines #1 and #2. In addition, statistics showed that the offspring of noncolored line #6 included fully colored, semicolored, noncolored, and dead seedlings, and the ratio of those four kinds of seedlings was close to 1:2:1:1. Moreover, it was found that two seedlings among the offspring of noncolored line #5 were colored (Figures 2f and S7). We speculated that the expression of *RUBY* might be unstable in different generations of *Arabidopsis*.

**Table 1.** Segregation of phenotypes in different lines of the $35S_{346\ bp}::RUBY$ transgenic *Arabidopsis* T2 generation.

| *35S::RUBY* Col-0 T2 | Total Number of Seedlings | Surviving Seedlings (%) | | | Dead Seedlings (%) | The Ratio of Surviving and Dead Seedlings | Chi-Test (*p*-Value) |
|---|---|---|---|---|---|---|---|
| | | Fully Colored Seedlings (%) | Semicolored Seedlings (%) | Noncolored Seedlings (%) | | | |
| #1 | 420 | 63.45 ± 2.99 | 11.44 ± 0.17 | 0.22 ± 0.22 | 24.89 ± 1.56 | 3.04 ± 0.43:1 | 0.97 |
| #2 | 676 | 73.31 ± 3.90 | 6.36 ± 1.01 | 0.00 ± 0.00 | 20.34 ± 2.81 | 4.09 ± 1.08:1 | 0.30 |
| #3 | 481 | 59.10 ± 1.88 | 19.10 ± 1.84 | 0.19 ± 0.19 | 21.64 ± 0.18 | 3.62 ± 0.07:1 | 0.44 |
| #4 | 445 | 0.00 ± 0.00 | 0.00 ± 0.00 | 75.13 ± 2.02 | 24.51 ± 2.36 | 3.08 ± 0.6:1 | 0.99 |
| #5 | 521 | 0.97 ± 0.37 | 0.00 ± 0.00 | 78.25 ± 1.15 | 20.77 ± 0.10 | 3.84 ± 0.39:1 | 0.80 |
| #6 | 676 | 17.51 ± 4.17 | 33.40 ± 5.15 | 26.78 ± 2.04 | 22.31 ± 1.60 | 3.53 ± 0.56:1 | 0.48 |

To further explore the relationship between *RUBY* expression stability and red coloration in heredity, two representative *RUBY* transgenic *Arabidopsis* lines, #3 and #6, were selected for further analysis. First, the RT−qPCR analysis showed that the expression levels of *CYP76AD1*, *DODA*, and *glucosyl transferase* in fully colored seedlings of the T2 generation of line #3 were the same as those of the T1 generation (Figure 6b). The expression levels of three genes in the T1 generation of line #6 were very low, but they were highly expressed in the T2 generation of fully colored seedling (Figure 6c). Moreover, we found that the expression levels of the three genes were very low in the semicolored and noncolored seedlings of the T2 generation of lines #3 and #6, and they showed no significant difference (Figure 6b,c). Second, we extracted betalain from the fully colored, semicolored, and noncolored seedlings in the T2 generation of lines #3 and #6. The results showed that the contents of betalain were consistent with RT−qPCR detection: the betalain contents of the fully colored seedlings were high, while the semicolored and noncolored seedlings had lower contents, and the difference between the latter two was not significant (Figure 6d,e). The above experiments further proved that the inconsistent red coloration in *RUBY* transgenic *Arabidopsis* was caused by its unstable expression. It is implied that the *RUBY* fusion gene might have transcriptional abnormalities in transgenic *Arabidopsis*, but it could also be due to RNA processing or other post-transcriptional regulation not known for *RUBY*.

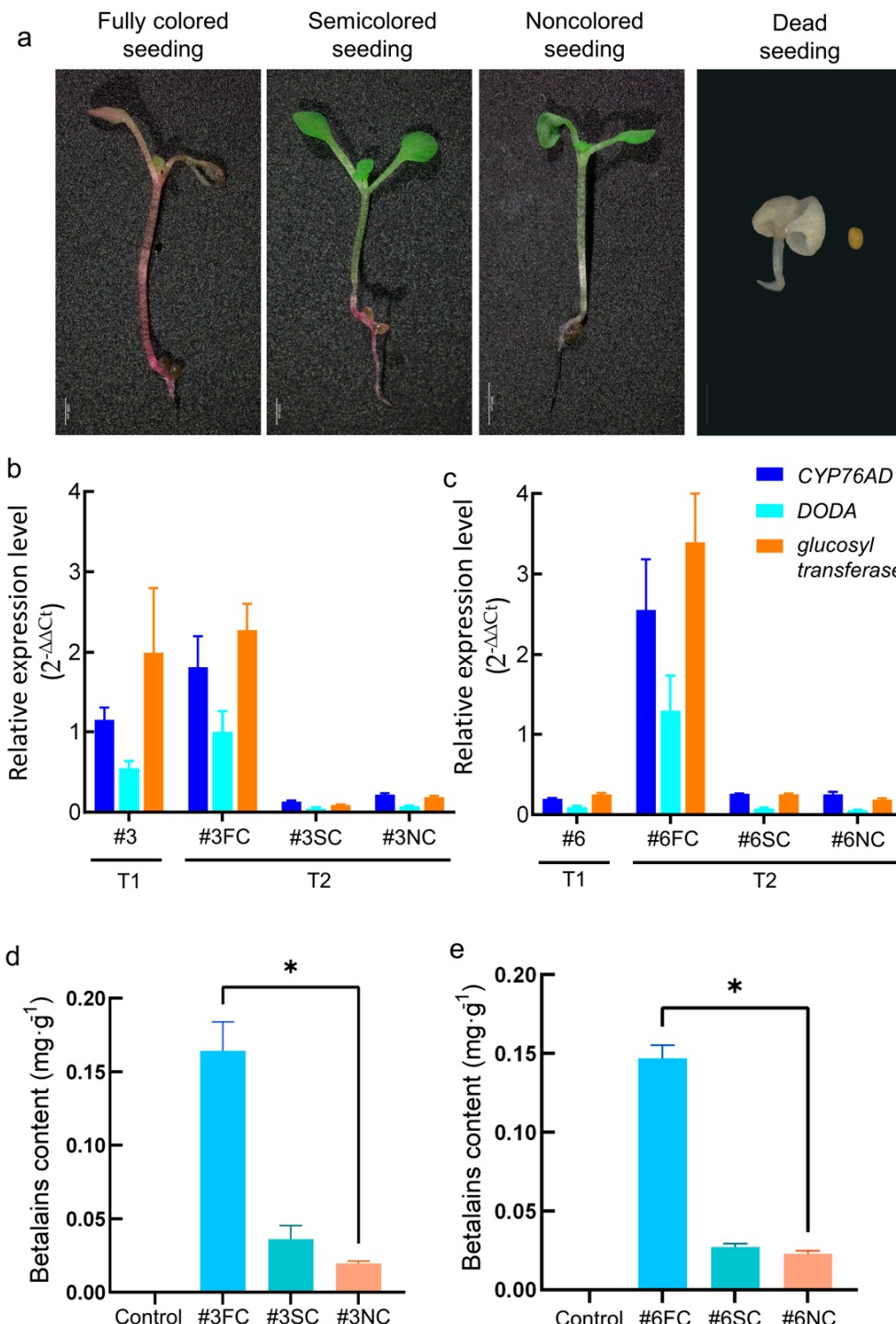

**Figure 6.** The unstable expression analysis of *RUBY* in transgenic *Arabidopsis*. (**a**) Different phenotypes of $35S_{346\ bp}$::*RUBY* transgenic *Arabidopsis* T2 generation (scale bars = 2000 μm). (**b**,**c**) RT−qPCR detection of *RUBY* gene expression of lines #3 and #6 T1 generation seedlings and fully colored, semicolored, and noncolored T2 generation seedlings. (**d**,**e**) Betalain content detection of lines #3 and #6 fully colored, semicolored, and noncolored T2 generation seedlings. WT seedlings were used as controls. * Indicates a significant difference in expression ($p < 0.05$).

## 4. Discussion

In this study, the reporter *RUBY* was used to reconstruct the betalain biosynthesis pathway in transgenic hairy roots of *P. volubilis* and in the model plants *Arabidopsis* and *N. benthamiana* to evaluate the potential applications of *RUBY* in plant genetic transformation.

### 4.1. RUBY Has Broad Prospects as a Naked-Eye Visual Screening Reporter

As a noninvasive reporter, *RUBY* is not only convenient for observation, but also does not affect callus induction, plant regeneration, or fertility. Generally, it has advantages in the genetic transformation of plants, allowing a nondestructive, easily visible evaluation of tissues without the need for fluorescence or light imaging, which can improve efficiency of genetic transformation and reduce the cost. Furthermore, due to the relatively simple synthetic pathway for betalains, *RUBY* has the potential to be applied on more plant species. Currently, the genetic transformation of woody plants is still relatively difficult [43]. The establishment of root transgenic systems mediated by *A. rhizogenes* has been widely used in studies of secondary metabolites and gene function. Therefore, the selection of an appropriate reporter is essential for the establishment of transgenic hairy roots [3,26,44,45]. Traditional selection reporters in genetic transformation, that generally use antibiotics and herbicides, have disadvantages for the growth of plant roots [46–48], and various fluorescent proteins or GUS require special equipment or expensive chemical substrates, which makes them inconvenient to observe [49,50]. Anthocyanin has been used as a naked-eye visual reporter, but its synthesis pathway involving at least eight genes is complex, and there were great differences between species [48,50–53]. Compared with the abovementioned reporters, *RUBY* is a novel, low-cost reporter that does not need instrument assistance and is an excellent alternative reporter in plant genetic transformation.

### 4.2. The Risk of RUBY in Genetic Transformation

As shown above (Supplementary Figure S1, Figures 2c and 4), a low rate of red coloration was observed in transgenic hairy roots of *P. volubilis* and *N. benthamiana* and *Arabidopsis* plants, in which the *CaMV 35S* promoter was used to drive the expression of *RUBY*. It was caused by variable expression of *RUBY* or instability of betalains. RT−qPCR analysis showed that the red coloration of *RUBY* was directly related to the level of expression. Although the *CaMV 35S* promoter has been widely used as a constitutive promoter to study gene function, there are differences in expression abundance in different tissues and abiotic stresses at each developmental stage of plants [54–56]. This was a good explanation for each of the regions that had different types of red coloration in the same transgenic hairy root of *P. volubilis*; white inflorescence appeared in fully colored transgenic *Arabidopsis* line #3, and small numbers of red seeds appeared in noncolored line #6, which might be caused by differential expression of *RUBY* driven by the *CaMV 35S* promoter. At the same time, the expression levels of *RUBY* in regions "*no.* 2" and "*no.* 5" were not significantly different (Figure 5c), but the coloration difference was obvious, probably due to the different accumulation of betalain in different tissues. It was reported that betalain accumulated differently in various tissues, affected by phytohormone and light conditions [57–59].

On the other hand, using a specific promoter to drive the expression of *CYP76AD1*, *DODA*, and *glucosyl transferase* in plants could enable the synthesis and accumulation of betalain in specific tissues [4,8]. However, we found that *CYP76AD1*, *DODA*, and *glucosyl transferase*, which constituted *RUBY*, were expressed at varying degrees in transgenic white hairy roots of *P. volubilis* and *N. benthamiana*, as well as the transgenic *Arabidopsis* lines and the white inflorescence of line #3. Relatively low expression of *RUBY* leads to insufficient accumulation of betalain and a noncolored plant, indicating that *RUBY* might have lower sensitivity than *GUS* and *LUC* [60,61], presenting the problem of false-negatives as a reporter. These results suggest that *RUBY* as a reporter requires a strong constitutive promoter to drive in transgenic plants, and a specific promoter or an inducible promoter used to drive *RUBY* may result in a reduced screening efficiency.

### 4.3. RUBY as a Reporter Needs Further Optimizations

Betalain has begun, in recent years, to be used as a reporter in studies on *Arabidopsis* and other species. However, we found that some tissues and plants with high expression of *RUBY* did not show enough red coloration. This may be a common problem with pigment-based reporters. For example, when *MdMYB10* was used as a reporter for promoter studies,

differences between the transgenic plants and the wild-type could hardly be distinguished in the case of a specific weak promoter, particularly in tissues showing a strongly colored background, such as green leaves. Even when using the 35S strong promoter to trigger plant coloration, not all the plants showed uniform coloration [62], and comparable results were shown in our study.

On the other hand, the accumulation of betalain depends on expression levels of biosynthetic genes and the stability of betalains. As a natural pigment, betalain is reported to have poor stability and is easily affected by factors such as pH, light, and temperature [63–65]. In this study, we compared the red coloration rates of *Arabidopsis* seedlings cultivated in a light–dark cycle and total darkness, and the red coloration of seedlings under different lighting conditions had no significant difference. However, this does not rule out the possibility that other factors we mentioned above affect betalain stability and accumulation as a reporter during the experiments, and further studies are needed to confirm this hypothesis.

The RNA expression levels of the three genes that make up the *RUBY* reporter gene system were variable in transgenic *P. volubilis* and *N. benthamiana* hairy roots, and transgenic *Arabidopsis* plants. The statistical data showed that pigment production and RNA expression over two generations of transgenic *Arabidopsis* were unstable. It is supposed that 2A peptide-mediated "self-cleavage" or ribosomal "skipping" to cleave multiple proteins does not always work efficiently [3]. There is more possibility: some unexpected post-transcriptional processes might happen in *RUBY* "polycistronic" mRNA for the three genes. Recent emerging evidence has shown that polycistronic mRNAs are also transcribed in eukaryotes. In *Arabidopsis*, 271 polycistronic loci have been identified and the post-transcriptional regulation for these loci remains to be elucidated [66]. Therefore, it is necessary to further fully understand the *RUBY* system and optimize it for wider use.

## 5. Conclusions

*RUBY* is a naked-eye visual reporter that has obvious advantages, such as a simpler synthesis mechanism, an easier experimental operation, and a lower cost in the genetic transformation of plants, and might be used as a high-quality reporter for the genetic engineering breeding of woody plants. However, the expression of this system was shown to be unstable in different generations of transgenic plants, which might be caused by post-transcriptional regulation. In order to achieve red coloration, the amount of accumulated betaines might need to reach a certain threshold. The sensitivity of *RUBY* system presents the problem of false-negatives as a reporter, but it still has broad potential application prospects after improvement.

**Supplementary Materials:** The following supporting information can be downloaded at https://www.mdpi.com/article/10.3390/f14030637/s1: Figure S1: Low red coloration of *RUBY* transgenic hairy roots of *P. volubilis* induced by *A. rhizogenes* K599. Figure S2: PCR identification of *RUBY* transgenic *Arabidopsis* lines. Figure S3: *RUBY* transgenic shoots of *N. benthamiana* induced by leaf-disk transformation methods. Figure S4: *RUBY* transgenic *N. benthamiana* plants grown in the pots. Figure S5: A nonuniform red coloration of *RUBY* transgenic hairy root of *P. volubilis.* Figure S6: Characterization of *RUBY* transgenic *Arabidopsis*. Figure S7: *RUBY* transgenic *Arabidopsis* offspring screening and red coloration.

**Author Contributions:** Conceptualization, Y.W. and Z.-F.X.; methodology, Y.W.; software, J.Y., S.D. and H.H.; validation, J.Y. and S.D.; formal analysis, J.Y.; investigation, J.Y., S.D., Y.W. and Z.-F.X.; resources, J.Y.; data curation, J.M.; writing—original draft preparation, J.Y. and Y.W.; writing—review and editing, Y.W. and Z.-F.X.; visualization, J.Y.; supervision, H.H. and J.M.; project administration, H.H. and J.M.; funding acquisition, J.Y., S.D. and H.H. All authors have read and agreed to the published version of the manuscript.

**Funding:** This work was supported by startup research funds from Guangxi University, Project of Guangxi Natural Science Foundation (2022GXNSFAA035491), and Key Research and Development Projects of Nanning (No. 20182005-1).

**Institutional Review Board Statement:** Not applicable.

**Informed Consent Statement:** Not applicable.

**Data Availability Statement:** Not applicable.

**Acknowledgments:** We would like to thank He Yubing of Nanjing Agricultural University for providing the *DR5::RUBY* vector, and Meng Dong of Beijing Forestry University for providing the *A. rhizogenes* K599 strain.

**Conflicts of Interest:** The authors declare no conflict of interest.

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
