# Peer review of "Exploring the Potential Applications of the Noninvasive Reporter Gene RUBY in Plant Genetic Transformation"

_forests, doi:10.3390/f14030637_

Round 1

Reviewer 1 Report

The article is devoted to the study of a new reporter gene with a view to its application in woody plants.

Despite the fact that a reliable technique for the transformation of woody plants has not yet been developed, the authors proposed an interesting model that allows quite simply use betalain to stain variants with successful transformation.

The goals set in the article have been achieved, applied methods are adequate.

The authors described in detail the materials and methods used, as well as the experimental conditions.

At the same time, the authors adequately perceive the obtained results and emphasize the limited possibility of the method for application in the transformation of woody plants.

Reviewer 2 Report

Forests Manuscript #:  2199041

Authors: J.-J. Yu et al.

Title:  Exploring the universality of the noninvasive reporter gene RUBY in plant genetic transformation

The authors have presented evidence for the use of the RUBY reporter gene to be used in transgenic plants as a visual, non-damaging (“noninvasive” – which can be interpreted to mean different things) screen for transgenic plants.  The RUBY reporter gene/construct is not new here, aand it introduces three genes/proteins needed to synthesize the red pigment betalain from the amino acid tyrosine.  Since tyrosine is present in all cells, it is a universal substrate for betalain pigment production.  The RUBY reporter gene construct these authors used is one that was previously published as being effective by another group, and properly cited here, with some modifications.  The previous published work with RUBY was in Arabidopsis and rice.  The authors in the manuscript under review transformed RUBY into (1) Arabidopsis, like the already cited previous work, along with (2) a woody dicot species Plukenetia volubilis (a vine crop plant used for oil/lipid production) and (3) the herbaceous dicot species Nicotiana benthamiana (a close relative to Tobacco and common model plant used for transgenic work and basic research in plants.  The goal of this study, as stated by the authors in both the title and abstract, is to test the universality and genetic and gene-expression stability of the RUBY reporter gene in a variety of plants.  They tested the visual production of betalain as well as RNA expression levels of the three genes/RNAs that make up the RUBY reporter gene system. They also tested genetic stability of pigment production and RNA expression over two generations of transgenic plants.  From these data, there are several challenges and issues that arise, as the authors point out.  The overall approach here is generally reasonable and the provided data are generally fine, but – as I see it - there are some questions and issues that need to be addressed.

My biggest scientific concerns with the manuscript are, firstly, that the implicit claim of this being a “universal” reporter gene, which the title alludes to, is not fully tested as I see it, when only three species (and all dicots) and very limited growth conditions were tested (mostly limited to tissue culture-grown conditions).  And, of the three species tested, only one is woody dicot (Plukenetia volubilis) that would seem to have direct relevance to the journal Forests.

Secondly, the authors state there was “no obvious side effects during plant growth” (in several locations in the paper).  However, most plants were only grown in highly protected tissue-culture conditions, and only limited number of plants were grown in soil and none in “field” settings, that I could see.  Furthermore, none of the growth conditions were in stressful environmental conditions. Thus, the claim of no obvious side effects is very limited. 

Finally, as the authors explained and based on the provided data, there are some very open questions as to variability and instability of expression and visible pigment production.  Thus, there remain a lot of questions about function / expression.  I do generally support the publication of this paper, after some of the questions listed below are resolved.

Suggested specific edits and/or questions, ordered chronologically by Sections, Paragraphs, and Lines (“L”) in manuscript:

Introduction:

Lines 33:  It seems clear the word “tree” is a typo and should be “three”, so as to read, “…pathways of the three: only CYP76AD, …”

Line 36:  Please define how the use of “noninvasive reporter” is being used here.  There are several different means for this, so the authors should explain their use of this term here.

Line 39:  Skipped the reference number “7”.  End of this line should be, “… endosperm [6,7].”  This then means the old “8” becomes new “7”, so forth for references and “old “9” becomes new “8”, and then the old “7” that first shows up in Line 46 becomes the new “9”.  The order of references in the bibliography would also be affected, as well as rest of citations for “old” 7, 8 and 9 through text.

Line 47:  Wording “cannot coexist” does not seem to fully fit this situation.  I would suggest instead, “…betelain and anthocyanins might be mutually exclusive.”

Line 48:  please clarify what is meant by saying, “…tomatoes can cover anthocyanin …”.  I believe I likely understand, but more clarity here would help.

Line 49:  I also feel a reference/citation is need for this section.  It might be reference #12, but that is not completely clear.

Lines 51 - 52:  More explanation of the somewhat unusual use of the “2A” proteins to allow production of three different proteins from a single gene.  Also, I strongly encourage the addition of the citation of reference #9 (old #) for Y. He et al., 2020 for it includes a good diagram of the RUBY reporter gene and explanation of the biochemical roles for the three proteins and where/how the very small “2A” protein included work.

Line 55:  The use of the term “selection” with transformation has very specific meaning that is different from how that word is being used here.  “Selection” with transformation is usually reserved for “selectable” markers that selected for growth / viability, such as antibiotic “selection”.  Here, the term “screen” would be fitting, for this is a visual “screen” for those that are transformed, as evidenced by the visible red pigment.

Lines 61:  Awkward wording of, “Those study provides…”. Please correct wording.

Line 70:  Same issue with “selection” when the term “screen” seems more fitting, as described above (Line 55).

Line 82:  More detail about how and where (greenhouse, field?) the soil-grown plants were kept until they bolted.

Lines 88 - 90:  A citation is needed here, in case it is different from “[23]” several sentences above.

Line 96:  An extra space is in “4 000”.  Seems it is missing the coma, “4,000”.

Lines 101-102:  Cite Table 1 so readers can see how many seedlings were included.

Line 121:  Formally, for the end of that line it would be a “reference RNA”, since RT-qPCR measures RNA abundance.

Lines 122-126:  This section of methods related to RT-qPCR method and quantification needs citations/references.  Also, what general type of qPCR being used should be stated. Was it a PacMan-based qPCR or SYBR green fluorescence.

Line 131:  Delete “in a centrifuge.”, for that is already mentioned earlier in that sentence.

Lines 134-135:  Please explain that the seedlings were classified by naked-eye visual inspection”, or some wording to that effect to make this clear, if that was indeed how they were classified.

Line 137:  extra, unneeded space is present between “red    pigment.”

Results:

Line 147:  Not clear what the “potential adverse effects on plant growth” would be from.  Are the authors suggesting that the presence of the DR5 promoter will have adverse effects on growth?  Please explain.

Line 156:  There were differences in the absorption spectra between red dragon fruit and RUBY Pvo callus in Figure 1e.  Namely, in the 400 – 460 nm range the RUBY Pvo has higher absorption than that of red dragon fruit.  Is this a range of different types of betalains or different pigments altogether?

Line 158:  I would not use “proved” for that is a very strong, and difficulty to know for certain with a single experiment, especially given the data in Fig 1e that shows different absorption spectra in the 400-460 nm range.  Perhaps, “which indicates that the pigments …. were like the same…”

Line 159:  I would provide a brief explanation of why red dragon fruit was used as a control for betalain pigments.

Figure 1 Legend:  For the absorption data, explanation of the statistics that is shown here.  How many replicates/samples (n = ?) that were tested?  Are these averages?  Needs error bars to show SD or SE.

Line 177:  The abbreviation ARM was defined in the methods, but it would be good to define it here again (Agrobacterium rhizogenes-mediate).

Lines 178:  As mentioned above in the opening section, the idea that plants grew normally when for some species they were mainly grown in tissue culture, makes it less convincing to accept the claim that they “grew without any abnormalities”.  To address this, I would suggest adding a statement such as, “under the growth conditions tested here.”

Line 180  add “Agrobacterium tumefaciens strain GV3101…”

Figure 2 Legend: “Harmless analysis” is awkward and not very clear wording.  Instead, perhaps, “Nondestructive” analysis might be better.

Line 243:  Delete, “In this study,” (its assumed the results are from this study) and instead start with, “We found that …”.

Lines 255-262:  This is a general observation/question.  It seems a bit surprising how what appears to e relatively small changes in RNA expression (Figure 5a) lead to what seems more dramatic betalain pigment production that is revealed in Figure 4a.  It as if there is threshold level of expression that is crossed that leads to a non-linear increase in betalain that then becomes visible.  My question is, is the result of real differences in betalain or more about our human-eye perception?

Figure 4:  In Figure 4d the first lane on the gel is labeled WT.  Persumably, the WT Arabidopsis is the Col-O plant/strain shown in Figure 4c.  But, this should be specified in the Legend and use same label in both the gel and plant picture panels, for clarity.

Lines 283-284:  I am not certain I agree there was higher expression in Pvo plant #2 (Pvo2) versus Pvo plant #5 (Pvo5), based on data provided in Figure 5a.  First, there were no p-value statistics for significance provided and there is not clear difference in expression between these two Pvo plants.  Yet, that is not was is stated in the text.  And, in Pvo5 the betalain red pigment is not observed.  Please include an explanation or discussion about this in the text.  Also, provide significance data for Figure 5a, somewhat similar to that tested/shown in Figure 5b.

Lines 297-298:  The wording of this last sentence is awkward, especially the last part, “…this problem was still unavoidable.”

Figure 5:  The “No Data” control included in all three panels is not a clear label.  As stated, it implies there were no data collected/obtained.  I suspect the authors mean to indicate that the data were below the threshold of detection or quantification, for how RT-qPCR is done.

Figure 5c:  The RT-qPCR data for the Nb1 – Nb9 plants is fine, but I did not see any data for the betalain pigment detection for each of these nine plants.  Supplemental Figure 1 has some Nb plants, but they are not numbered such that one can tell which is Nb1, Nb2, etc…, to related to the Figure 5c expression data.

Figure 5:  General question and concern regarding the RNA expression data.  Since all three “RNAs” (CYP76AD1, DODA, and glucosyl transferase” quantified here are all from the same, single “polycistronic” mRNA, based on how the RUBY reporter gene with the “2A” protein system works, then why would there be any difference in RNA level between these three?  And, for some plants there are dramatic differences (Pvo3, Pvo4, At1, At3, Nb4, Nb7, etc…).  This is where a Northern blot might reveal more information to know if in fact there is just one “polycistronic” mRNA for all three proteins OR is there some unexpected post-transcriptional processes happening here?  Please provide some explanation as to why this might be.

Line 310:  Write out words for numbers less than 10, so, “three” in both locations on this line.

Lines 311-314:  What is the cause of the “dead” seedlings?  This is very unclear.  And, if fitting Mendel’s law of inheritance for  3:1 ratio, that would suggest single gene with standard dominant/recessive.

Line 334:  Expression that is “nearly 1 times” is the same as saying, “same expression”, which is more clear.

Lines 341-344:  The authors mention the unstable expression might be due to transcriptional abnormalities.  That might be true, but the authors also cannot rule out that it might also be due to RNA processing or other post-transcriptional regulation not know for the RUBY reporter gene.

Figure 6:  The dashed rectangle shown in both 6b and 6c are not explained. Please add explanation in the Figure Legend.

Discussion:

Line 356: As mentioned above, these are visual “screens” and not selections, a term usually reserved for the selection that blocks / prevents growth or viability, such as antibiotic selection, etc…

Line 359:  A repeat of what the advantages are would be helpful here, especially in light of the instability and variable expression issues revealed in this manuscript.

Line 363: the reference “67” is a typo, and it should be reference “37”.

Line 368:  Write out word for “eight.”

Line 425:  The word “mature” at end of sentence does not seem the correct word to use here.  Perhaps instead, “not sufficiently understood and optimized to be fully useful.”

Line 427: Again, what are advantages?

Line 429: It is not clear what the fusion gene “design defects” are here.  Is it the problems with the variable expression?  That does not necessarily mean it is a “design defect”, but instead might just be the variability in the different tissues, cells, etc…

Reviewer 3 Report

The authors present a detailed study of both the applicability of the RUBY reporter construct in Plukenetia volubilis transformation and its stability in this and other species. The evaluation of color and gene expression in different tissues and generations allows for a broad comparison of the efficiency of the reporter in generating the betalain reporter. These experiments successfully show the instability of the RUBY construct and the risks of false interpretations associated with its use. Further experiments to evaluate transcript and protein stability would be interesting to confirm the authors' findings.

General comments:

The Introduction is good, but it would benefit from more information regarding using RUBY in other species. It has been successfully used in several species, and other works also discuss the universality of this reporter. Please include more information regarding the conclusions other researchers have reached.

The Results are interesting and well-presented. However, the authors could improve them to remove redundancy and clarify the information.

The Discussion would benefit from less redundant information and a deeper exploration of the discussed topics. More dense use of the literature is essential to enhance the Discussion. The importance of a non-invasive reporter in genetic transformation of tree species can be better highlighted.

Minor comments and suggestions:

  1. Line 15: This system has been used in different species, so be careful not to disregard previous works.

  2. Line 33: "...pathway of the tree" -> "...pathway of the three"

  3. Line 50: Please cite reference [9] at the end of the first phrase.

  4. Line 57: Please use examples of RUBY in other species to provide more information on its universality and stability. These articles describe the use of RUBY in other species: Phyllostachys edulis: https://doi.org/10.1007/s12374-020-09294-y#Sec2 ; Ailanthus altissima: https://doi.org/10.1016/j.xinn.2022.100345 ;Persea americana: https://doi.org/10.1007/s11240-022-02436-9 ;Ipomoea batatas: https://doi.org/10.1101/2023.01.02.522521

  5. Line 61: "Those study provides" - > "Those studies provide"

  6. Line 62: “hard-to-transform” -> “recalcitrant

  7. Line 62: "" to verify the universality" -> "to assess the universality"

  8. Lines 65-66: "obtained through the mediation of A. rhizogenes," -> "obtained by A. rhizogenes mediated transformation"

  9. Lines 66-71: This phrase is too long. Breaking it in two would make the flow of the text better. I suggest breaking the phrase in line 69 between "Arabidopsis, which provided" -> "Arabidopsis. These experiments provided"

  10. Line 84: “The plasmid of pRN114” -> “The pRN114 plasmid”

  11. Line 94: "plus" -> "with"

  12. Line 95: "into LB liquid medium": The authors gave specific volumes in the previous phrases. Please make the style consistent and either (a) provide specific quantities for every step of the protocol or (b) remove specific quantities for every step.

  13. Line 103: "method in soybean with some" -> "method in with some"

  14. Line 112: Table 1 is not in the correct place. Please check.

  15. Lines 120-121: "Actin and its homologs…" -> "Actin homologs…"

  16. Lines 122-124: It is unclear why biological replicates were used in Arabidopsis and tobacco, and only technical replicates were used for P. volubilis.

  17. Line 141: What the authors mean by universality needs to be clarified. Some published papers show that the system is not effective universally. The authors explore the system's usefulness to this specific species and under control of the specific studied promoter. A better phrasing would be "To explore the potential applications of RUBY as a reporter in the genetic…", since this is the first woody plant where RUBY was used.

  18. Line 143: This result is expected since DR5 is indeed affected by auxin, as described in the next sentence. Additionally, try to use concrete numbers or percentages instead of "some" and "rest". For example, "Of 100 observations, 20 were red, and 80 were white."

  19. Line 150: "such as NEOMYCIN PHOSPHOTRANSFERASE II (NPT II)" - this information is unnecessary.

  20. Line 158: Please cite references, if present, regarding the use of dragonfruit to study betalains.

  21. Lines 160-164: This last paragraph needs to be clarified... The information is not easily understandable because it seems to be the same given at the beginning of the first paragraph.

  22. Line 188: It would be interesting to compare tyrosine quantification between ruby plants and wild type to show whether it is being replenished.

  23. Line 210: Please clarify whether the transgenic Arabidopsis were generated with A. rhizogenes or otherwise. Since the paragraph begins discussing A. rhizogenes, it induces the reader to think that the analyzed plants were generated by A. rhizogenes transformation.

  24. Lines 215-224: Using percentages to describe the number of red roots would make the information easier to understand.

  25. Line 280: The word "basically" is too informal and unnecessary here. Please remove.

  26. Line 300: The results presented in Figure 5 complement Figure 4. It would be beneficial to present these results together. The figures of the different plants (Figure 4 a, c, e, f) could come as supplementary material.

  27. Lines 309-310: Please clarify that the T2 seedlings are Arabidopsis T2 seedlings.

  28. Lines 322-324: This phrase is redundant.

  29. Line 331: The word "basically" is too informal.

  30. Line 335: Since the RUBY transcript encodes all three genes, how could a single transcript have different abundances depending on its region? Are there biological explanations for this phenomenon? Could there be differences in the efficiency of the PCR primers or other technical explanations? Which experiments could be performed to assess this? Please discuss this thoroughly in the Discussion.

  31. Lines 349-350: “semicolor” - > “semicolored”

  32. Line 353: The word "novel" is unnecessary here.

  33. Line 374: Remove "still"

  34. Line 393: Please mention how these risks could be evaluated.

  35. Line 404: "same results" -> "comparable results"

  36. Line 409:" and it turned out that" is too informal. Use "and"

  37. Line 412: Some of the candidate factors could be mentioned here.

  38. Lines 420-425: I suggest the authors explore the epigenetic and transcriptional possibilities further before jumping into protein cleavage. Also, discuss whether MycoRef was better or worse than RUBY. It is unclear.

  39. Line 429: The experimental evidence is insufficient to prove that the instability results from design defects in the fusion gene. Please revise this phrase.
